# A Non-Invasive, Label-Free Method for Examining Tardigrade Anatomy Using Holotomography

**DOI:** 10.3390/tomography11030034

**Published:** 2025-03-14

**Authors:** Minh-Triet Hong, Giyoung Lee, Young-Tae Chang

**Affiliations:** 1Department of Chemistry, Pohang University of Science and Technology, Pohang 37673, Republic of Korea; atonidas@postech.ac.kr; 2Molecular Imaging Center, Pohang University of Science and Technology, Pohang 37673, Republic of Korea; dirldudrldl@postech.ac.kr

**Keywords:** holotomography, tardigrade anatomy, label-free imaging

## Abstract

Background/Objectives: Holotomography is an advanced imaging technique that enables high-resolution, three-dimensional visualization of microscopic specimens without the need for fixation or staining. Here we aim to apply holotomography technology to image live *Hypsibius exemplaris* in their native state, avoiding invasive sample preparation procedures and phototoxic effects associated with other imaging modalities. Methods: We use a low concentration of 7% ethanol for tardigrade sedation and sample preparation. Holotomographic images were obtained and reconstructed using the Tomocube HT-X1 system, enabling high-resolution visualization of tardigrade anatomical structures. Results: We captured detailed, label-free holotomography images of both external and internal structures of tardigrade, including the digestive tract, brain, ovary, claws, salivary glands, and musculature. Conclusions: Our findings highlight holotomography as a complementary high-resolution imaging modality that effectively addresses the challenges faced with traditional imaging techniques in tardigrade research.

## 1. Introduction

Holotomography is an advanced imaging technique that enables high-resolution, three-dimensional visualization of microscopic specimens without the need for labels or dyes [1,2]. This innovative method integrates the principles of quantitative phase imaging and optical diffraction tomography to reconstruct the three-dimensional refractive index distribution within a sample. By capturing multiple two-dimensional holographic images from various angles of illumination and computationally reconstructing them into a comprehensive three-dimensional image, holotomography achieves unparalleled imaging clarity. The technique is non-invasive and minimizes phototoxicity, making it particularly well suited for imaging live specimens in their native states. Its capability to visualize the intricate anatomy of biological samples has opened up new frontiers in biological and biomedical research, enabling researchers to investigate complex biological systems without altering or damaging the specimens. Recent advancements in instrumentation, such as the development of the Tomocube system, have made holotomography more versatile and accessible [3]. These advancements ensure that even non-expert researchers can readily adopt this technique for a wide range of applications, further broadening its impact in the scientific community.

Tardigrades, or water bears, are renowned for their extraordinary ability to survive extreme environmental conditions through anhydrobiosis, entering a dehydrated, ametabolic “tun” state [4]. Unlike other desiccation-tolerant organisms that rely on trehalose, tardigrades employ unique molecular adaptations, notably tardigrade-specific intrinsically disordered proteins (TDPs), which vitrify into a protective glass-like matrix upon drying [5]. Other protective proteins, such as CAHS and LEA, contribute to their ability to endure extremes of temperature, radiation, and pressure, allowing them to revive upon rehydration [6]. Beyond passive protection, tardigrades actively repair damage, particularly radiation-induced DNA breaks, through a robust DNA repair system and a unique damage suppressor protein, Dsup, which shields DNA from fragmentation [7,8,9]. Tardigrade proteins have been shown to stabilize sensitive therapeutics like Factor VIII, potentially eliminating cold storage requirements for biologic drugs and vaccines [10]. Their resilience also informs astrobiology, as their survival in space and resistance to cosmic radiation provide insights into life’s potential for interplanetary transfer [11]. Developmentally, tardigrades exhibit a simplified body plan with a reduced Hox gene set, suggesting extreme evolutionary truncation from segmented ancestors [12]. Their embryogenesis follows direct development, revealing fundamental and streamlined developmental mechanisms applicable to broader evolutionary studies. Additionally, tardigrade-derived biomolecules inspire innovations in biomimetic materials, including stress-responsive gels and vitrification-based stabilizers that could protect cells, enzymes, or pharmaceuticals in fluctuating environments [13]. Altogether, tardigrades have profound implications for biotechnology, including regenerative medicine, biopharmaceutical preservation and astrobiology.

Various imaging modalities have been developed to provide valuable insights into tardigrades’ intricate anatomy. Scanning electron microscopy (SEM) has revealed the simultaneous formation of body segments, limb buds, and claws [14], as well as intricate details of stylet structures [15]. Transmission electron microscopy (TEM) has further elucidated the subcellular architecture of tardigrades, including structural changes during dehydration [16]. However, both SEM and TEM require extensive sample fixation and preparation, which are time-consuming and can introduce artifacts, thereby altering the natural state of the specimens.

Fluorescent confocal microscopy has also been employed to explore key neural components, such as the brain and segmental trunk ganglia [17,18], and has elucidated the processes underlying germ cell development into mature oocytes [19]. Additionally, immunostaining techniques have revealed segment-specific gene expression patterns that underlie body patterning [20]. While fluorescent imaging is a powerful tool for these investigations, its effectiveness is hindered by several challenges, including limited dye penetration, intricate sample preparation, and autofluorescence from structures such as stylets, claws, algae, and birefringent granules in the midgut. Additionally, immunofluorescence imaging is affected by photobleaching and phototoxicity [21], which can further reduce the specificity of fluorescence staining and imaging while also compromising cellular viability.

Holotomography, a label-free, non-invasive imaging modality that exploits intrinsic refractive index differences, has proven highly effective for imaging live, physiological organisms without staining or fixation. Studies on zebrafish embryos and larvae have demonstrated that refractive index holotomography can reconstruct 3D images of key anatomical features in vivo, often enhanced by pigment inhibitors and index-matching media [22,23]. Similarly, tomographic phase microscopy and optical diffraction tomography on Caenorhabditis elegans has achieved sub-micron resolution imaging of internal organs such as the pharynx and digestive tract; the use of non-toxic RI-matching agents like iodixanol further enhances the contrast of fine details [24,25,26]. Overall, holotomography enables superior visualization of living organisms compared to traditional imaging modalities that require invasive sample preparation (Appendix A).

In this study, we, for the first time, present holotomography on live *Hypsibius exemplaris* using the Tomocube HT-X1. This non-invasive, rapid, and efficient approach eliminates the need for fixation and staining, allowing us to image live tardigrades in their natural state without compromising their viability. By simply anesthetizing the tardigrades with a low concentration of ethanol, we are able to obtain high-resolution, three-dimensional images of both external and internal structures—including the digestive tract, brain, ovary, claws, salivary glands, and musculature—without the need for invasive procedures or fluorescent labeling. Our findings demonstrate that holotomography is a powerful tool for detailed, label-free, and non-phototoxic imaging of live tardigrades, enabling rapid and comprehensive analysis of their anatomy and organ systems.

## 2. Materials and Methods

### 2.1. Tardigrade Culture

*Hypsibius exemplaris* (catalog number 133960, Carolina Biological Supply, Burlington, NC, USA) were cultured in 60 mm plastic Petri dishes maintained at a constant temperature of 25 °C. To enhance locomotion, the bottom surfaces of the dishes were lightly abraded with P-100 sandpaper, providing necessary traction for the tardigrades. The culture medium consisted of dechlorinated tap water, which was prepared by allowing the water to stand for at least 24 to eliminate chlorine and chloramine, which are toxic to tardigrades. Tardigrade cultures typically thrive in dechlorinated tap water or natural spring water rather than ultra-pure water, as the trace minerals and appropriate pH levels in these water sources support their health. In contrast, pure deionized water may lack essential ions necessary for their well-being. The water was refreshed weekly to maintain optimal water quality and to remove waste products. Tardigrades were fed weekly with Chlorella sp. algae (catalog number 152069, Carolina Biological Supply, Burlington, NC, USA), ensuring a consistent food supply.

For optimal results, it is crucial to ensure thorough dechlorination of the tap water used in the culture medium [27]. Failure to adequately dechlorinate the water can result in chlorine or chloramine toxicity, adversely affecting tardigrade health and culture stability. Regular monitoring of water quality and timely replacement of the culture medium are essential practices to maintain a robust and healthy tardigrade culture.

Maintaining a stable temperature of 25 °C is critical for the health and activity of *Hypsibius exemplaris*. Deviations from this optimal temperature can adversely affect tardigrade physiology and behavior. Therefore, it is essential to monitor and regulate the incubation environment carefully to ensure temperature consistency, thereby promoting robust and viable cultures for experimental applications.

### 2.2. Quantification of Anesthetization and Recovery Durations in Tardigrades

*Hypsibius exemplaris* were anesthetized using a freshly prepared 7% (*v*/*v*) ethanol solution, which was created by diluting absolute ethanol (≥99.5%) with tap water immediately before each experiment to ensure consistency and prevent concentration shifts due to evaporation. All procedures were conducted at 25 °C. Tardigrades were gently transferred using a micropipette into a watch glass containing the 7% ethanol solution. The immobilization process was observed under an inverted microscope, with the time from initial ethanol exposure until complete cessation of movement recorded for each specimen. Post-immobilization, tardigrades could remain in the 7% ethanol solution for up to 3 h without exhibiting adverse effects.

Anesthetized tardigrades were carefully transferred back into distilled water to facilitate recovery, substituting tap water to eliminate any residual ethanol. Recovery time was documented from the moment of transfer until each tardigrade resumed normal motility, including leg movements, body flexion, and feeding behavior. To assess long-term viability, tardigrades were returned to their standard culture plates after recovery, and their behavior and survival were monitored over several days to confirm normal activity. Key considerations for replication include maintaining a stable temperature of 25 °C during both anesthetization and recovery phases to ensure optimal tardigrade health, preparing the 7% ethanol solution immediately prior to each experiment to prevent concentration variations due to ethanol evaporation, and using the same water source consistently throughout the procedure, unless explicitly comparing different conditions, to avoid introducing variability.

### 2.3. Tardigrade Sample Mounting for Holotomography Imaging

Following anesthetization, individual *Hypsibius exemplaris* were carefully transferred to individual wells of a six-well plate (P06-1.5H-N, Cellvis, Mountain View, CA, USA). A precise 4 µL droplet of 7% ethanol in tap water was placed at the center of each well, ensuring that the tardigrade remained fully immersed within the droplet. A custom-sized coverslip (4.2 × 4.2 mm), cut from clear plastic paper, was put onto a small, raised platform fitted within the well. By gently lowering the plate, the droplet made contact with the coverslip, and the perimeter of the coverslip was sealed with clear nail polish to minimize evaporation. Care was taken to avoid trapping air bubbles, as they can interfere with refractive index calculations during imaging.

For holotomography imaging using the Tomocube HT-X1 (Tomocube, Daejeon, Republic of Korea), it is critical that the tardigrade is precisely positioned at the center of the well within the 4.2 × 4.2 mm square field of view. Since the Tomocube HT-X1 cannot visualize samples outside of this region, accurate placement ensures a streamline imaging process. 

### 2.4. Brightfield Microscopy

For brightfield imaging, tardigrade images were captured using an Axiocam 506 mono camera (Carl Zeiss Microscopy, Jena, Germany) paired with a 40× objective lens (420762-9900-000, Carl Zeiss Microscopy, White Plains, NY, USA) and illuminated by a Colibri 7 LED light source (Carl Zeiss Microscopy, Jena, Germany) on a Zeiss Axio Observer microscope (Carl Zeiss Microscopy, Jena, Germany).

### 2.5. Holotomography Using Tomocube HT-X1

Holotomography imaging was conducted using a Tomocube HT-X1 system equipped with a 40× objective lens (numerical aperture ~0.95). The expected refractive index (RI) range was set to 1.33 for water and tardigrade tissues, acknowledging that certain structures, such as claws and stylets, may slightly exceed this range. Each three-dimensional dataset was acquired over approximately 10 s, with the system automatically rotating illumination angles during this period. Reconstruction of the acquired data was performed using TomoAnalysis v2.0 (Tomocube, Daejeon, Republic of Korea), resulting in a final voxel resolution of approximately 156 nm in the xy plane and 714 nm along the z-axis (Appendix A). The instrument’s chamber was maintained at 25 °C with the use of an additional heated stage. All reconstructions were carried out using TomoAnalysis v2.0, employing the default reconstruction algorithm without additional post-processing beyond the built-in filtering. 

### 2.6. Data Analysis and Visualization

Brightfield images were processed using ImageJ version 1.54j (NIH, Madison, WI, USA). For holotomography data, three-dimensional reconstructions were examined and sectioned using TomoAnalysis v2.0. Pseudo-color schemes were applied to represent depth, facilitating the differentiation of anatomical structures. Z-stacks of the RI maps were saved in TIFF format for archival purposes and subsequent analysis.

## 3. Results

### 3.1. Tardigrade 3D Holotomography

Previous research has demonstrated that tardigrades possess a high tolerance to alcohol, surviving for up to 7 days in high alcohol concentrations [28]. In our study, we observed that tardigrades can be effectively anesthetized using a low concentration of 7% ethanol in distilled water. Immobilization was achieved within 5.2 ± 1.9 min (mean ± standard deviation, *n* = 7) of incubation in this solution (Figure 1). Moreover, tardigrades remained anesthetized in 7% ethanol for up to 24 h (Figure 1 and Appendix A). After that, tardigrades were transferred back to dechlorinated tap water, regaining consciousness and reflexes within 7.7 ± 2.4 min and 10.9 ± 5.6 min (mean ± standard deviation, *n* = 7) after removal from 3 h and 24 h incubation, respectively, with 7% ethanol solution as the anesthetic effects gradually diminished (Appendix A). All specimens resumed normal locomotion and feeding behavior. These findings indicate that 7% ethanol does not adversely affect *Hypsibius exemplaris* viability.

After anesthetization with 7% ethanol, individual *Hypsibius exemplaris* were placed in a 4 µL droplet at the center of a six-well plate and immobilized under a 4.2 × 4.2 mm coverslip (Figure 2a). The droplet, containing the tardigrade, was then covered and sealed with the coverslip to prevent ethanol evaporation and subsequent changes in ethanol concentration, ensuring that the anesthetized specimen remained fully immobilized during holotomography imaging. Each tardigrade remained stationary throughout the multi-angle data acquisition, which required approximately 10 s per 3D dataset. All holotomography experiments were conducted at 25 °C to maintain consistent physiological conditions and prevent ethanol evaporation or condensation, which could otherwise alter the refractive index (RI) during imaging. Notably, no significant twitching or body movements were observed during Tomocube imaging following anesthetization with 7% ethanol.

Since tardigrade body lengths in our samples ranged from approximately 150 µm to 250 µm, encompassing both juveniles and adults [29], we utilized the tile imaging mode of the Tomocube HT-X1 system to ensure comprehensive coverage of the entire tardigrade during imaging. The Tomocube HT-X1 performed LED-based multi-angle illumination, yielding a series of 2D holographic images that were computationally merged via the built-in TomoAnalysis (Appendix A). The resulting 3D reconstructions encompassed the entire body volume of each tardigrade, from dorsal to ventral surfaces (Figure 2b).

All reconstructed volumes were exported as TIFF-formatted z-stacks and as 3D objects for interactive rendering in TomoAnalysis (Figure 2b,c and Appendix A). The resulting three-dimensional holotomography image can be visualized using different pseudo-color or color gradient profiles to indicate depth within the three-dimensional environment (Figure 2b). Additionally, the holotomography image can be viewed as a z-stack comprising multiple layers of the holotomogram (Figure 2c). Each tardigrade’s z-stack consisted of approximately 50–60 slices along the z-axis (depending on tardigrade thickness), with each slice spaced 714 nm apart.

### 3.2. Anatomy of Tardigrade Visualized Using Holotomography

A major advantage of holotomography over conventional light microscopy is its ability to generate high-resolution, three-dimensional (3D) images of both external and internal tardigrade structures in a single, label-free experiment (Figure 3). In *Hypsibius exemplaris*, the external cuticle, head region, trunk segments, and legs are readily observed at shallow z-planes, whereas deeper layers—such as the digestive tract, reproductive organs, salivary glands, and musculature—emerge in subsequent focal planes. By toggling through z-stacks, researchers can isolate individual organs in near-native states without the optical overlap that commonly obscures such features under brightfield conditions.

#### 3.2.1. External Morphology

At a shallow focal plane (z ≈ 0 µm; Figure 3b, upper panel), holotomography distinctly captures the head (he), trunk segments (ts1–4), and each pair of legs (le1–4). The tardigrade often appears as a defined boundary with a higher refractive index than the surrounding medium (RI of water ~1.33) (Figure 3b upper panel). This clear demarcation enables visualization of fine surface features such as body folds and indentations that are difficult to resolve in brightfield images (Figure 3a). Each leg terminates in claws (cl), specialized cuticular structures secreted by claw glands (cg). Holotomography can differentiate the claw’s unique refractive index from that of soft tissues, making these relatively small structures stand out clearly in 3D reconstructions.

#### 3.2.2. Feeding Apparatus and Digestive Tract

When focusing more deeply within the tardigrade body (z ≈ 20 µm; Figure 3b, lower panel and Appendix A), the mouth (mo) region becomes more evident (Figure 3b, lower panel, and Appendix A), along with the dual calcareous stylets (st) that tardigrades use to pierce food sources [30]. These stylets lead into the buccal tube (bt), a channel connected to a muscular pharynx (ph). The pharynx is responsible for generating suction during feeding and thus appears thicker and more refractive compared to the narrower buccal tube.

Beyond the pharynx lies the esophagus (oe), which delivers ingested material to the midgut (mg). Tardigrade midguts often contain varying amounts of ingested food and storage cells (sc)—free-floating cells that store energy reserves (e.g., glycogen, proteins, lipids) [31]. In individuals with extensive storage cells, the hindgut may be partially obscured. However, holotomography’s optical sectioning capability can reveal the hindgut’s continuity and boundary in tardigrades with fewer storage cells (Figure 3c and Appendix A).

#### 3.2.3. Reproductive Organs

Holotomography clearly reveals the ovary (ov), which is situated dorsally near the midgut (Figure 3b lower panel and Appendix A). This structure is responsible for oocyte production and can be identified by its distinct refractive properties relative to surrounding tissues.

#### 3.2.4. Salivary Glands and Musculature

Another feature uniquely resolved by holotomography is the positioning of salivary glands (sg) near the pharynx (Figure 3c and Appendix A). These glands release enzymes aiding in the pre-digestion of food. Because they exhibit a slightly different refractive index than surrounding tissues, holotomography allows them to be distinguished even without fluorescent labeling.

Musculature (mu) in tardigrades typically consists of bands of muscle fibers attached to the inner cuticle, controlling limb and body movements. Holotomography can capture these muscle fibers, particularly in specimens with fewer storage cells where internal tissues are less crowded (Figure 3c and Appendix A). Under brightfield microscopy, muscle bands are often obscured by overlapping tissue layers. In contrast, holotomography’s optical slicing separates them into distinct planes, enabling detailed examination of muscle arrangement, thickness, and attachment points.

Overall, holotomography surpasses conventional light microscopy in capturing the three-dimensional organization of tardigrade tissues. By mapping both external structures (cuticle, claws) and internal organs (digestive tract, salivary glands, musculature, ovary). This label-free method offers an unprecedented view of tardigrade anatomy in near-native conditions, enabling more accurate, comprehensive analyses of these resilient micro-animals.

## 4. Discussion

Here, we, for the first time, perform holotomography on a live tardigrade *Hypsibius exemplaris*. We are able to visualize detailed three-dimensional images of both external and internal structures, including the digestive tract, brain, ovary, claws, salivary glands, and musculature, in a label-free and non-phototoxic manner. By anesthetizing the tardigrades with a low concentration of 7% ethanol, we effectively immobilized them without affecting viability, enabling high-clarity imaging that surpasses conventional light microscopy. Our findings demonstrate that holotomography is a powerful tool for comprehensive visualization of tardigrade morphology and organ systems, revealing intricate details on live tardigrade.

We present a non-invasive, rapid, and efficient approach for examining live tardigrade anatomy with high-resolution imaging. This method eliminates the need for time-consuming and invasive sample fixation typically required in electron imaging techniques. Electron imaging on tardigrades requires complete fixation with chemical fixatives such as glutaraldehyde and osmium tetroxide, followed by a series of dehydration steps using increasing concentrations of ethanol [32]. The specimens are then embedded in resin, sectioned, and often stained to enhance contrast before imaging. This lengthy preparation is not only time-consuming but also involves harsh chemicals that can introduce artifacts and alter the natural ultrastructure of the tissues. Moreover, these processes render the tardigrades non-viable, making it impossible to study dynamic physiological processes in living organisms. In contrast, our holotomography-based approach involves minimal sample preparation and allows for the imaging of live specimens in their native state.

Furthermore, holotomography offers a label-free, non-phototoxic alternative to fluorescent imaging for studying tardigrades. While fluorescent imaging provides valuable insights, its effectiveness is limited by challenges such as dye penetration and depth-related signal loss. Furthermore, autofluorescence from structures like stylets, claws, algae, and birefringent granules in the midgut can reduce the specificity of fluorescence staining and imaging.

Holotomography could offer a transformative approach to studying developmental biology and embryogenesis in tardigrades. A previous study showed that in *Eimeria bovis*, this label-free imaging technique successfully tracked real-time morphological changes throughout sporogony—from the early sporoblast/sporont stages to the final formation of sporocysts and sporozoites—without relying on fixation or staining [33]. By capturing sequential 3D images, researchers were able to characterize developmental sub-phases and quantify key cellular parameters such as compactness, area, and dry mass. Translating these capabilities to tardigrade systems would allow detailed time-lapse monitoring of organogenesis, body segment formation, and cell division dynamics in near-native conditions. Furthermore, the same digital staining and morphometric analyses that helped distinguish sporogony stages in *E. bovis* would enable comparative studies of different tardigrade species or genetic variants. Ultimately, holotomography not only enriches our understanding of tardigrade developmental processes but also facilitates rapid, quantitative evaluations of potential stressors or interventions—mirroring its proven utility in parasitological research.

Holotomography also holds significant potential in pharmacological and toxicological screening using tardigrades. Another study showed that label-free, real-time observation of drug-induced morphological changes via holotomography is feasible for red blood cells, where 3D and 4D refractive index tomograms were used to track both concentration- and time-dependent membrane alterations [34]. Continuous holotomographic monitoring of tardigrades could uncover how chemicals—including heavy metals, pharmaceuticals, or other environmental toxins—diffuse and interact with tissues and organs, revealing dose-dependent disruptions without the need for fluorescent labels. This approach thus enables rapid toxicity evaluations and drug screening in an alternative, resilient invertebrate model, expanding the scope of holotomography in biomedical and environmental research.

## 5. Conclusions

In summary, we demonstrate the successful application of holotomography for high-resolution, label-free imaging of live *Hypsibius exemplaris*. This innovative method enabled detailed three-dimensional visualization of both external and internal structures, including the digestive tract, brain, musculature, and reproductive organs, without compromising the specimen’s viability. Compared to conventional techniques like electron microscopy and fluorescence imaging, holotomography offers significant advantages, including non-invasiveness, minimal sample preparation, and avoidance of phototoxicity or labeling artifacts. By utilizing 7% ethanol as an effective yet gentle anesthetic, we ensured precise imaging in the native state of tardigrades. These findings highlight holotomography as a powerful tool for advancing the study of tardigrade biology and for broader applications in non-invasive, live animal imaging in biological research.

Holotomography represents a transformative approach for future study of tardigrades. Leveraging its ability to capture sequential, high-resolution images without fixation or staining, future research could employ this technique for detailed time-lapse monitoring of embryogenesis, organogenesis, and cellular dynamics, thereby elucidating the intricacies of body segmentation and developmental processes. Moreover, its sensitivity to subtle refractive index variations opens up promising avenues for pharmacological and toxicological screening, allowing real-time investigation of how chemicals, heavy metals, and environmental toxins interact with tardigrade tissues. Given the unique molecular adaptations that underpin tardigrade resilience, holotomography could also facilitate comparative studies of genetic variants and stress responses, ultimately advancing applications in regenerative medicine, biopharmaceutical preservation, and astrobiology.

## Figures and Tables

**Figure 1 tomography-11-00034-f001:**
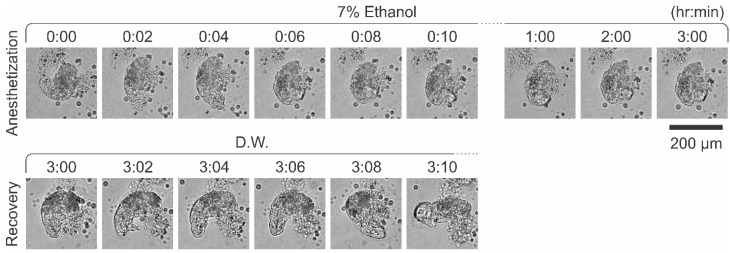
Anesthetization of tardigrades using 7% ethanol in distilled water. Tardigrades were exposed to 7% ethanol in distilled water, and images were captured at various time points. After 3 h or 24 h, the tardigrades were transferred to a new well containing distilled water, and imaging continued at different time points. Scale bar = 200 µm.

**Figure 2 tomography-11-00034-f002:**
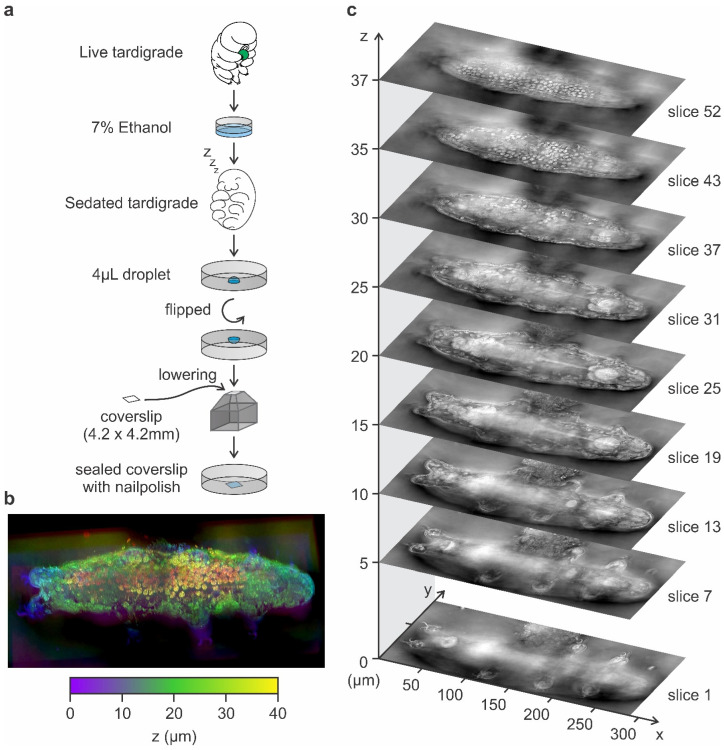
Tardigrade 3D holotomography. (**a**) Schematic of tardigrade preparation for holotomography imaging. Tardigrades are sedated in 7% ethanol in tap water and placed in a 4 µL droplet at the center of a 6-well plate. A 4.2 × 4.2 mm coverslip is carefully positioned by lowering the plate until the droplet comes into contact with and adheres to the coverslip. The coverslip is sealed with nail polish to prevent evaporation during imaging. (**b**) Three-dimensional reconstruction derived from the holotomography, presented with a pseudo-gradient color scale (purple–green–yellow) to indicate image depth (0 μm–20 μm–40 μm). (**c**). Representative slices from the reconstructed holotomography stack, showcasing a z-resolution of 714 nm and an xy-resolution of 156 nm (Appendix A).

**Figure 3 tomography-11-00034-f003:**
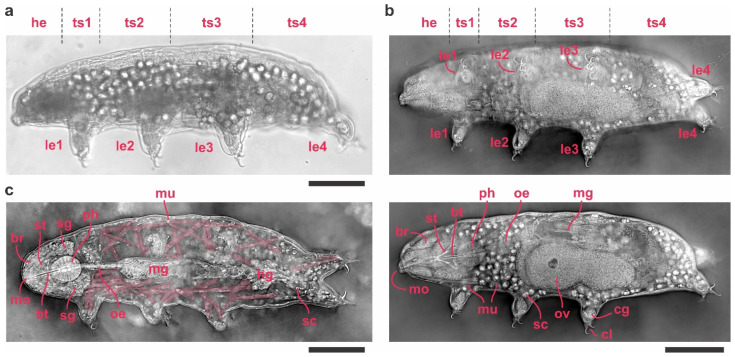
Anatomy of tardigrade visualized using holotomography. (**a**) Brightfield image of a tardigrade. (**b**) Reconstructed holotomogram at two different z-planes, with the upper panel highlighting the external morphology and the lower panel revealing internal anatomy, including the digestive tract, ovary, and others. (**c**) Reconstructed holotomogram illustrating musculature, highlighted in light red. Scale bar: 50 µm. Abbreviations: he, head; ts1–4, trunk segments 1–4; le1–4, legs 1–4; br—brain; st—stylet; bt—buccal tube; ph—pharynx; oe—oesophagus; mg—midgut; mg—hindgut; mo—mouth; mu—muscle; sc—storage cells; sg—saliva glands; ov—ovary; cl—claws; cg—claw glands.

## Data Availability

The source data for the current study has been uploaded and are available via the Figshare repository at the following link: https://doi.org/10.6084/m9.figshare.28518710.v1 (accessed on 11 March 2025).

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
