# Peer review of "A Non-Invasive, Label-Free Method for Examining Tardigrade Anatomy Using Holotomography"

_tomography, 2025, doi:10.3390/tomography11030034_

Round 1

Reviewer 1 Report

Comments and Suggestions for Authors

Thanks for the opportunity to review this article. In an overall view, this is quite interesting research, with well-designed figures and impressive findings. I list several personal concerns that I believe are helpful to make this article better.

  • In the introduction part, 2nd paragraph, what area can tardigrade research contribute to? Could the authors extend the introduction? The readers, including myself, are really curious about what can we learn using tardigrade.
  • As a novel imaging technique, a schematic figure should be added to illustrate the imaging mechanism of holotomography.
  • The performances of holotomography are impressive, I would recommend the authors to provide some schematic figures or photographs of the imaging system, which can aid readers to further learn about the technical details.
  • Please list the vital performance indicators of holotomography (sensitivity, resolution, 3D field of view, etc.) using a table.
  • Other than tardigrades, are there any types of living animals suitable for in vivo holotomography?
  • The authors provided several tomographic images with depth, could these slices become 3D rendered?
  • Multiple organs can be visualized under holotomography, is it possible to colorize each organs within pseudo-colors?

Author Response

Dear Reviewer 1, please see the attachment.

Reviewer 2 Report

Comments and Suggestions for Authors

The manuscript is of high quality and is nearly ready for publication, pending minor revisions on the following points:

  1. "Including limited dye penetration, intricate sample preparation, and autofluorescence from structures such as stylets, claws, algae, and birefringent granules in the midgut, which can compromise the specificity of fluorescence staining and imaging." In addition to these limitations, photobleaching and phototoxicity are important considerations in immunofluorescence. Highlighting these issues further emphasizes the advantages of holotomography.

  2. The authors should briefly explain why they used dechlorinated tap water instead of the more commonly used double-distilled or deionized water. Additionally, they should clarify how they ensured the quality of dechlorination.

  3. The authors should justify their use of nail polish to seal the sample instead of a more commonly used anti-fading sealant. Additionally, the source and composition of the nail polish should be specified.

  4. The font in sections 3.2.1 to 3.2.4 differs from the rest of the text and appears blurry, making it difficult to read. This should be standardized for clarity and readability.

Author Response

Dear Reviewer 2, please see the attachment.

Reviewer 3 Report

Comments and Suggestions for Authors

Summary:
This work illustrates a protocol for 3D optical imaging of very small live whole organisms. The authors use holotomography which is an optical method that synthesizes views collected in a circular pattern around a target to synthesize a tomographic series with contrast based on refractive index. Some advantages of this approach are that it is fast, does not involve high-power lasers, and does not require fixation nor labeling. The authors examine tardigrades which are well understood by a variety of microscopy methods and thus an interesting choice to illustrate a new application of holotomography. The authors explain that they were able to sedate the animals with a moderate concentration of ethanol during imaging. Animals were then allowed to recover in tap water and apparently this was successful. The images produced are clearly of high quality and complement what can be seen in standard bright field microscopy. In holotomography the muscle groups are especially visible compared to unstained bright field microscopy. Some comparison is made to other microscopy methods, but only one optical bright field image is shown for comparison.

Opinion:
This is mostly an application note and I think not much is new or discovered in the paper, but it is still an interesting contribution for this relatively young method of holotomography.

Strengths:
Convincing imaging of the animals. Nice figures with high quality of presentation and descriptive captions. Decent amount of background literature is cited.

Weaknesses:
Original data not available except by request.
No visual comparison to other microscopy methods like electron microscopy, confocal microscopy with labeling, dark field microscopy, etc. This might be interesting for the reader's understanding. I think it would be quite find to request to reproduce other figures and there is not a need to create new microscopy images.
Conclusion does not explain gaps in our understanding or very well suggest future directions

Areas for Improvement:
Source data should be provided in a microscopy repository (Image Data Repository [IDR], Cell Image Library, Zenodo, Dryad, Figshare, MorphoSource, etc.). I am immediately aware of one similar example which is on Zenodo: https://zenodo.org/record/1250729. In my experience (see also http://www.pnas.org/cgi/doi/10.1073/pnas.1708290115), asking authors for their data does not usually result in me receiving the data. This kind of friction is a barrier to open science and with all these free and low cost tools to share your data I find this to be a requirement for publication. There is not any obvious intellectual property or ethical reason to not share these data, so I hope this request can be honored.
It is explained that this method could be used for long term imaging experiments, but this is not really tested in this paper. The authors kept the animals in 7% ethanol for 3 hours and then returned them to 0% ethanol (100% tap water). I don't think they imaged through that 3 hour period and I am not sure that 3 hours is long enough to address the kinds of experiments proposed on lines 311-317. I would like some notion of how feasible it would be to do these experiments with either comments from the authors or citations to other studies with examples of how long live imaging experiments take to study these things. In my view, this is the weakest claim in the paper.
Please include a version number (if available) for the TomoAnalysis software. That way, the work is more clearly reproducible. Especially since the meaning of phrases like 'employing the default reconstruction algorithm' might change if the makers of this software change that method.

Line Item changes:
94: change 'remained' to 'remain'

References to consider:
http://www.pnas.org/cgi/doi/10.1073/pnas.1708290115
https://zenodo.org/record/1250729

Author Response

Dear Reviewer 3, please see the attachment.

Round 2

Reviewer 1 Report

Comments and Suggestions for Authors

I have no further concerns to this article.

Author Response

Reviewer: I have no further concerns to this article.

Answer: We deeply appreciate the time and effort you invested in assessing our work, and we are grateful for your constructive insights. Your feedback has been invaluable in enhancing the quality of our article, and we are pleased that no further concerns remain.

Reviewer 3 Report

Comments and Suggestions for Authors

The authors graciously provided an updated version of their manuscript that includes revisions based on comments by myself and, I assume, other reviewers. All points that I raised were directly addressed by the authors. I am very pleased with all of the revisions. The additional figure for supplementary material nicely captures some differences in approach to imaging these animals. I am also delighted that the authors have said they will upload their original images to a repository. Unfortunately, the link is not working for me as of today (March 3, 2025), but as long as this is done and the page is made public I think it is ok. Several smaller edits were also incorporated. One of the most complete author responses I've received when reviewing.

Author Response

Reviewer: The authors graciously provided an updated version of their manuscript that includes revisions based on comments by myself and, I assume, other reviewers. All points that I raised were directly addressed by the authors. I am very pleased with all of the revisions. The additional figure for supplementary material nicely captures some differences in approach to imaging these animals. I am also delighted that the authors have said they will upload their original images to a repository. Unfortunately, the link is not working for me as of today (March 3, 2025), but as long as this is done and the page is made public I think it is ok. Several smaller edits were also incorporated. One of the most complete author responses I've received when reviewing.

Answer: We sincerely appreciate your thorough and constructive review of our revised manuscript. Your positive feedback on the revisions, including the additional supplementary figure and the uploaded source data, is greatly valued. We have updated the manuscript to include the corrected link (https://doi.org/10.6084/m9.figshare.28518710.v1) and have verified that the source data link is fully functional. Thank you for recognizing the comprehensiveness of our responses and for your invaluable insights, which have significantly contributed to improving the quality of our work.